# Exploratory Analysis of Circulating Serum miR-197-3p, miR-1236, and miR-1271 Expression in Early Breast Cancer

**DOI:** 10.3390/ijms26188944

**Published:** 2025-09-14

**Authors:** Burak İlhan, Sibel Kuraş, Berkay Kılıç, Ceren Tilgen Yasasever, Hilal Oğuz Soydinç, Hani Alsaadoni, Gözde Öztan, Arash Adamnejad Ghafour, Muhammed Ucuncu, Enver Kunduz, Süleyman Bademler

**Affiliations:** 1Department of Surgery, Istanbul Faculty of Medicine, Istanbul University, 34093 Istanbul, Türkiye; burak.ilhan@istanbul.edu.tr; 2Department of Medical Biochemistry, Hamidiye Faculty of Medicine, University of Health Sciences, 34668 Istanbul, Türkiye; sibel.kuras@sbu.edu.tr; 3Department of Surgery, Oncology Institute, Istanbul University, 34093 Istanbul, Türkiye; berkaykilic28@yahoo.com; 4Department of Basic Oncology, Oncology Institute, Istanbul University, 34093 Istanbul, Türkiye; ceren.yasasever@istanbul.edu.tr (C.T.Y.); hoguz@istanbul.edu.tr (H.O.S.); 5Department of Medical Biology, International School of Medicine, University of Health Sciences, 34668 Istanbul, Türkiye; hani.alsaadoni@sbu.edu.tr; 6Department of Medical Biology, Istanbul Faculty of Medicine, Istanbul University, 34093 Istanbul, Türkiye; gozdeoztan@istanbul.edu.tr; 7Department of Cancer Genetics, Oncology Institute, Istanbul University, 34093 Istanbul, Türkiye; arash.adamnezhad@gmail.com; 8Department of Anesthesia, Vocational School of Health Services, Istanbul Gelisim University, 34310 Istanbul, Türkiye; muhammeducuncu@gmail.com; 9Department of Surgery, Faculty of Medicine, Bezmialem Foundation University, 34093 Istanbul, Türkiye; drkunduz@yahoo.com

**Keywords:** breast cancer, microRNA, biomarker, early diagnosis, circulating miRNA, qRT-PCR, liquid biopsy

## Abstract

Early detection of breast cancer (BC) remains a challenge despite advances in screening, leading to poor prognosis in late-stage disease. Circulating microRNAs (miRNAs) have emerged as promising non-invasive biomarkers for early diagnosis. The study aimed to evaluate the diagnostic utility of serum miR-197-3p, miR-1236, and miR-1271 levels in distinguishing BC patients from healthy individuals. Serum samples from 92 BC patients and 31 healthy controls were analyzed. Total RNA was extracted, and miRNA expression levels were quantified using quantitative real-time PCR (qRT-PCR). Expression differences were assessed using the ΔCt method. Receiver Operating Characteristic (ROC) curve analysis evaluated diagnostic performance. Serum miR-197-3p levels were significantly upregulated (fold change = 8.939, *p* = 0.0048), while miR-1236 was downregulated (fold change = 0.112, *p* = 0.0029) in BC patients. miR-1271 showed no significant association. miR-1236 demonstrated superior diagnostic performance (AUC = 0.731, sensitivity = 80.7%, specificity = 51.3%) compared to miR-197-3p (AUC = 0.667, sensitivity = 67.0%, specificity = 45.6%). The combination of miR-197-3p and miR-1236 improved diagnostic accuracy (AUC = 0.842, CI: 0.764–0.936). Serum miR-197-3p and miR-1236 hold promise as complementary biomarkers for early BC detection. Larger multicenter studies are warranted to validate their clinical utility.

## 1. Introduction

Breast cancer (BC) is the most frequently diagnosed cancer and a leading cause of cancer-related mortality among women worldwide, accounting for over 2 million new cases annually. Notably, it has a cancer-related mortality rate of 7%, with 43,000 deaths in 2023 [1,2]. Early-stage BC is curable; however, delayed diagnosis often results in poor clinical outcomes due to metastasis and therapeutic resistance [3]. Although the latest diagnostic and therapeutic options are improving daily, long-term survival outcomes can be unfavorable. Early diagnosis of BC is critical. Therefore, there is an urgent need for the adjunctive use of biomarkers that are both sufficiently sensitive diagnostic tools and can be adapted to effective treatment strategies. Several lifestyle and reproductive factors have been associated with the development of this type of cancer, including early menstruation, delayed menopause, postponed childbirth, shortened breastfeeding durations, post-menopausal hormone replacement therapy, oral contraceptive use, alcohol consumption, and obesity [4]. The underlying cause of 5–10% of BC is considered to be hereditary gene mutations, especially BRCA1 and BRCA2 [5]. BC development is influenced by both intrinsic factors within cancer epithelial cells and the tumor microenvironment [5,6]. Mammary stem cells (MaSCs) are present as a very small proportion of undifferentiated cells in the mammary gland and can generate new MaSCs through self-renewal as well as give rise to a variety of differentiated cells through symmetric and asymmetric divisions. It is assumed that asymmetric divisions give rise to progenitor cells. However, their distinction from cancer stem cells has not been established due to their similar effects in tumorigenesis and tissue regeneration through various interactions in the tumor microenvironment [7]. Many signaling pathways have been elucidated in breast tumorigenesis. Gene expression profiling techniques have unveiled a diverse spectrum of gene dysregulations in BC samples, underscoring the intricate genomic architecture of BC [8]. In this regard, besides BRCA mutations, the PI3K (Phosphatidyl-inositol 3-kinase)/AKT/mTOR (the mammalian target of rapamycin) pathway, the ATM gene, TP53 gene, and Li-Fraumeni syndrome, PTEN (Phosphatase & Tensin homolog) and Cowden syndrome, STK11 or LKB1 and Peutz-Jeghers syndrome, PALB2 (Partner and Localizer of BRCA2), Checkpoint kinase 2, and CDH1 germline mutations are the most frequently detected BC oncogenic alterations [5].

In another area, epigenetic alterations, particularly aberrant DNA methylation and histone modifications, have garnered substantial recognition and are increasingly employed in cancer research. These enzymatic processes that govern the epigenome present promising avenues for developing therapeutic strategies to address transcriptional abnormalities inherent to the cancer epigenome. Consequently, most mammalian transcripts are noncoding RNAs, which regulate gene expression through diverse mechanisms [9]. Noncoding RNAs (ncRNAs) are classified based on their size and functional characteristics. Long noncoding RNAs (lncRNAs) are ncRNAs with a length exceeding 200 nucleotides, whereas ncRNAs with a length less than 200 nucleotides encompass microRNAs (miRNAs), small nucleolar RNAs (snoRNAs), and piwi RNAs (piRNAs). miRNAs are short RNA molecules (typically 20–24 nucleotides) that do not undergo protein translation. Originating from DNA transcription, they play a pivotal role in post-transcriptional gene regulation by binding to the 3′ untranslated regions (3′ UTR) of target messenger RNAs (mRNAs) [10]. miRNAs exert a pivotal role in regulating both transcriptional and post-transcriptional gene expression by establishing specific interactions with target mRNAs [11]. miRNA dysregulation and altered expression levels are implicated in various carcinogenesis processes, with either the tumor suppressor or oncogenic roles [12,13,14], drug resistance [15], or play roles in the pathogenesis of other diseases, including inflammatory diseases [16,17].

miR-197-3p is a small ncRNA molecule of length 22 nucleotides. Usually, it binds to the 3′ UTR of target mRNAs, reducing their stability or inhibiting their translation. This feature allows miR-197-3p to exert a strong regulatory effect on gene expression. It controls many biological processes at the genetic and epigenetic levels and is, therefore, associated with cancer development. Overexpression of miR-197-3p in BC cells increases cell proliferation. However, this miRNA can inhibit apoptosis in cancer cells [18].

Many studies have observed that miR-1236 is lowly expressed in BC cells, and that this low expression is associated with increased tumor growth and metastasis. In particular, miR-1236 has been shown to inhibit the ability of cells to invade and migrate. These properties suggest that miR-1236 has a tumor suppressor effect in BC cells. In one study, low expression of miR-1236 suppressed target genes that regulate the cell cycle and affect apoptosis pathways [19]. miR-1236 can be used as a biomarker as it shows low expression even in the early stages of cancer. Studies have shown that miR-1236 showed low expression in the serum of BC patients, and that these levels are associated with the stage of cancer. That suggests that miR-1236 can be critical as a biomarker in clinical diagnosis and prognosis [20].

miR-1271 is bound to the 3′ UTR of mTOR mRNA, thereby inhibiting its translation. Low expression of miR-1271 was associated with increased proliferation and cellular growth in BC cells via SPIN1. miR-1271 is thought to be a target of some genes that suppress cell cycle regulation and cellular growth. Suppression of miR-1271 may lead to faster division of cancer cells and tumor growth. The inhibition of apoptosis in BC cells and the low expression of miR-1271 may also be associated. That may result in cancer cells surviving and developing resistance to treatment [21].

Considering these multifaceted effects of miRNAs, this study aimed to investigate the levels and expression characteristics of miR-197-3p, miR-1236, and miR-1271 in serum samples as a noninvasive approach for early diagnosis, which may be needed and critical in BC.

## 2. Results

The median age of all participants, including 92 patients and 31 controls in our study, was 51.2 years. In contrast, the median age of the patient group was 54.2 years, while that of the control group was 47.8 years. Serum ΔCt levels of miR-197-3p and miR-1236 are correlated with each other (*p* < 0.001, r = 0.591), and with BC (miR-197-3p, *p* = 0.002, r = −0.272, miR-1236, *p* = 0.001, r = 0.284). Notably, miR-1271, which is correlated with other miRNAs, is not associated with the disease. Serum miR-197-3p and miR-1236 ΔCt values and fold change coefficients calculated for the associated expression changes were statistically significant with the diagnosis of BC. According to these values, miR-197-3p exhibited an upregulation with a fold change of 8.939, while miR-1236 showed a downregulation with a fold change of 0.112. However, no disease-associated regulation data were also found for miR-1271. Serum ΔCt values and fold change data are shown in Table 1 and Figure 1.

Of the participants, 50.4% were premenopausal, and 49.6% were postmenopausal. Notably, 28 (22.8%) participants had not given birth, while the remaining 72 (77.2%) had given birth at least once. Furthermore, 20 participants (16.3%) had a history of potentially life-threatening or malignancy-transforming vices such as smoking, alcohol, or substance abuse, while the remaining 103 (83.7%) participants did not. Nineteen (15.4%) participants had used oral contraceptives before or after childbearing, regardless of their birth status. Conversely, the remaining 104 (84.6%) participants had no history of oral contraceptive use. We observed that miRNA ΔCt values did not change in the entire participant group, depending on these factors specific to these individuals. ΔCt values of miR-197-3p and miR-1236 stratified by menopausal status, birth history, smoking/alcohol habits, and oral contraceptive use. No statistically significant differences were observed (*p* > 0.05 for each comparison).

While miR-197-3p and miR-1236 were found to be effective according to univariate analysis, these miRNAs were also effective according to multivariate analysis, with miR-1236 being the most effective one according to (forward) logistic regression analysis (OR: 5.06, *p* < 0.001). There was no discernible difference in age (OR: 1.80, *p* = 0.164) or menopausal status (OR: 2.17, *p* = 0.072) between the patient and control groups in the participants. Participants of the patient group exhibited a significantly higher prevalence of at least one birth. In contrast, the control group demonstrated a more pronounced prevalence of bad habits and a history of oral contraceptives. The distribution of these variables between patients and controls is shown in Table 2.

Regarding their significance in early BC diagnosis, miR-1236 was more effective than miR-197-3p with an area under the curve (AUC) of 0.731 versus 0.667, respectively. In this model, the positive and negative predictive values for miR-1236 were 89.3% and 53.7% with the cut-off value of 9.87, while the values for miR-197-3p were 79.6% and 30.9% with the cut-off value of 7.32. In addition, in the model created by determining the reference cut-off values of miRNAs, we found that the AUC reached up to 0.842 (CI: 0.764–0.936) when we used these miRNAs in combination (Figure 2, Table 3 and Table 4).

For miR-197-3p and miR-1236, which have significant ΔCt values, we did not detect any predictive or prognostic properties in the BC patient group (Table 5) (*p* > 0.05 for all comparisons).

Since the number of BC patients with proven gene mutations was minimal, any analysis that could be associated with miRNA levels was not possible. In addition, we did not find any correlation between miRNA levels and serum protein levels such as CA15-3 or CEA, with *p* > 0.05.

## 3. Discussion

Since improving disease-related survival and reducing mortality are considered to be the most important parameters, cancer biomarkers are becoming increasingly valuable in early diagnosis, prediction of the disease’s prognostic behavior, and evaluation of response or non-response to treatment options. In this regard, microRNAs (miRNAs) are small RNA molecules that regulate gene expression at the post-transcriptional level and play critical roles in the inflammatory processes, drug resistance, or the pathogenesis of many cancer types, including BC. Recent studies [8,9,10,11,12,13,14,15,16,17,18] have helped us better understand the role of miRNAs in BC biology and have shown that these molecules can be used as potential biomarkers and therapeutic targets in diagnosis, prognosis, and treatment strategies. Recent studies have also examined the role of miRNAs in BC progression through immune-molecular mechanisms. It has been demonstrated that miRNAs offer promising immunotherapy options for BC in order that they interact with numerous immunological factors in the tumor microenvironment and regulate immune cells in the immune response [22].

miRNAs may contribute to BC development by regulating various cellular processes. For example, miRNAs such as miR-21, miR-155, and miR-210 can promote proliferation, invasion, and tumor metastasis [23]. However, some miRNAs may have dual functions. miR-335 has both oncogenic and tumor suppressor properties. Expression of miR-335 can modulate the proliferation and invasion of tumor cells by affecting genes such as BRCA1, ER-α, and IGF1R [24]. miRNAs stand out as potential biomarkers in the early diagnosis and prognosis assessment of BC. In particular, miRNAs such as miR-195 and miR-210 are effective in tumor stages and metastatic potential [25]. miRNAs may affect the response to BC treatment. In particular, miR-205 may contribute to treatment resistance by regulating proteins such as MED1 and HER3. Low miR-205 expression may lead to high expression of MED1 and HER3, promoting the development of treatment resistance [24]. In this regard, this study investigated the diagnostic utility and potential roles in tumor progression of identified circulating serum miR-197-3p, miR-1236, and miR-1271 in BC.

miR-197-3p plays an important role in regulating fundamental biological processes, particularly cellular proliferation, apoptosis, migration, and invasion. The post-transcriptional regulation of this miRNA on its target genes can be decisive in modulating cellular responses. miR-197-3p has recently attracted attention as a microRNA that plays critical roles in the pathogenesis of various pathological conditions such as cancer, cardiovascular diseases, and autoinflammatory disorders. Regarding cardiovascular diseases, it has been reported that miR-197-3p induces damage in human coronary arterial endothelial cells by regulating target genes such as TIMP3 and IGF1R, especially in Kawasaki disease [26]. On the other hand, a study suggested that LINC00664/hsa-miR-197-3p/JAK2 interaction may be a biomarker for life-threatening myocardial fibrosis after acute myocardial infarction [27]. miR-197-3p also plays a role in the pathogenesis of autoinflammatory diseases. In patients with Familial Mediterranean Fever (FMF), miR-197-3p has been reported to modulate inflammation in monocytes and synovial fibroblasts by targeting the IL1R1 gene. This finding suggests that miR-197-3p may play roles in regulating immune responses and may be a potential treatment target of autoinflammatory diseases [28]. miR-197-3p may contribute to tumor progression by modulating processes such as proliferation, invasion, and resistance to therapy in cancer cells. In osteosarcoma, miR-197-3p has been shown to increase cancer stem cellularity and resistance to chemotherapy by inhibiting the SPOPL gene [29]. It has also been reported to promote angiogenesis in lung adenocarcinoma metastasis by targeting TIMP2/3 through cancer-derived exosomes, and overexpression of miR-197-3p may promote carcinogenesis [30,31]. Furthermore, the study showed that the miR-197-3p/API5 axis triggers progression and radioresistance in glioblastoma [32]. In contrast, miR-197-3p has been shown to act as a tumor suppressor in prostate cancer by regulating the VDAC1/AKT/β-catenin Signaling Axis [33]. The effect of miR-197-3p in BC plays an important role in regulating key cancer processes, particularly cell proliferation, invasion, metastasis, and resistance to therapy. Studies in BC cells show that the expression of miR-197-3p is usually high [34]. This increased expression may promote cancer cell growth and invasion into surrounding tissues. Li et al. reported that miR-197-3p is overexpressed in BC cell lines such as T47D or MCF7 cells, especially in tamoxifen-resistant cell lines. That suggests that mir-197-3p plays a critical role in carcinogenesis and tamoxifen resistance. In the same study, HIPK3 was found to be down-regulated in BC tissue, negatively correlated with miR-197-3p, and consequently promoted carcinogenesis by targeting HIPK3 via miR-197-3p [34]. According to the study of Ye et al., circFBXW7 has the activity of sponging miR-197-3p. This sponging effect decreases the expression levels of FBXW7 protein, which acts as a tumor suppressor, as a result of miR-197-3p overexpression and promotes carcinogenesis, tumor progression, especially in triple-negative BC [35]. In our current study, serum miR-197-3p ΔCt values correlated with BC diagnosis (miR-197-3p, *p* = 0.002, r = −0.272). The fold change coefficient calculated for the associated expression changes was statistically significant with the diagnosis of BC. According to these values, miR-197-3p exhibited an upregulation with a fold change of 8.939.

Data on miR-1236-related studies generally emphasize its tumor suppressor role and drug resistance properties. In lung adenocarcinoma, miR-1236 negatively correlates with Zinc finger E-box binding homeobox 1 (ZEB1). ZEB1 promotes epithelial–mesenchymal transition (EMT). Therefore, miR-1236 had the opposite effect by reversing ZEB1’s promoting effect on EMT. In addition, it has been reported that miR-1236 inhibition and ANKRD22 overexpression increase cisplatin resistance in treatment, and this axis can be used as a therapeutic target [36,37]. Song et al. reported that decreased expression of miR-1236 via regulation of miR1236-3p/TRIM37 axis plays a role in the promotion effect of cervical cancer [38]. It has also been demonstrated that the reduction in miR-1236 suppression effect by targeting MTA2 expression promotes gastric cancer progression [39]. It was found that alpha-fetoprotein (AFP) expression increased in HBV-associated cancer cells due to hepatitis B virus infection, whereas miR-1236 expression decreased, and this could be a therapeutic approach by regulating AFP expression [40]. Regarding the relationship between miR-1236 and BC, Yu et al. demonstrated that circ_0102273 and 6-phosphofructo-2-kinase/fructose-2, 6-biphosphatase 3 (PFKFB3) protein showed high expression in BC cells and inversely correlated with miR-1236, and confirmed that miR-1236 was sponged by circ_0102273. Therefore, they showed that controlling the regulation of circ_0102273 by targeting the miR-1236-3p/PFKFB3 axis may be a critical therapeutic option that can inhibit proliferation and metastasis in BC [41]. Huang et al. showed that inhibiting the effect of circADAM9, which works with similar expression relationships with miR-1236 and fibroblast growth factor 7 (FGF7), in preventing BC progression via the miR-1236-3p/FGF7 axis may be a good target for treatment [42]. Targeting the inhibition of circSPECC1, which has similar expression relationships, through the miR-1236/CBX8 axis has also emerged as an anti-BC drug [43]. Hao et al. showed that circ_0006528 and chromodomain helicase DNA-binding protein 4 (CHD4) were upregulated, and miR-1236, which is sponged by circ_0006528, was downregulated in BC cells showing resistance to adriamycin, a particularly important adjuvant chemotherapy agent. They also showed that this mechanism promotes cancer cell proliferation and invasion. They suggested that reversing this mechanism may be a step in reducing drug resistance [44]. In our current study, serum miR-1236 ΔCt values correlated with BC diagnosis (miR-1236, *p* = 0.001, r = 0.284). The fold change coefficient calculated for the associated expression changes was statistically significant with the diagnosis of BC. According to these values, miR-1236 exhibited a downregulation with a fold change of 0.112. Notably, neither miR-197-3p nor miR-1236 expression was influenced by patient characteristics such as menopausal status, reproductive history, smoking, or oral contraceptive use. Additionally, these miRNAs did not correlate with tumor size, stage, or receptor status, suggesting that they may reflect intrinsic tumor biology rather than clinical features. Logistic regression confirmed miR-1236 as an independent predictive marker (OR = 5.06, *p* < 0.001).

In the literature, studies show that miR-1271 is significantly down-regulated in BC tissues and affects BC progression through various pathways [45]. Although previously implicated in mTOR signaling and treatment resistance for various cancers, miR-1271 did not show diagnostic relevance in our cohort, potentially due to tumor subtype, heterogeneity, or its role in later disease stages.

Again, some studies in the literature provide valuable information on the diagnostic potential of various miRNAs, such as miR-210, miR-16, miR-21, and miR-195 alone or used in combination, for the diagnosis of BC and predict that they will contribute to the early diagnosis of BC as a non-invasive method [23,24,25,26]. Zablon et al. reported a review that many other miRNAs, including miR-1246, miR-1307-3p, miR-4634, miR-6861-5p, and miR-6875-5p, can be used with high specificity and sensitivity for early diagnosis [24]. Fontana et al. showed that the miR-200 family was down-regulated in aggressive BC subtypes such as Luminal B, Her2, and triple negative [46]. In our study, we emphasized miR-197-3p and miR-1236, which are relatively less studied as diagnostic biomarkers. According to this present report, while both miRNAs demonstrated individual diagnostic potential (AUC = 0.667 for miR-197-3p and AUC = 0.731 for miR-1236), the combined analysis yielded significantly improved accuracy (AUC = 0.842), supporting the use of multi-marker strategies in early cancer detection. In addition, a meta-analysis based on examining a large group of miRNAs revealed that the diagnostic success of miR-155 was superior to mammography. Although the patient group in our study consisted of early-stage BC patients, we did not make any comparisons since all patients had cancer findings on imaging. The same meta-analysis suggests that combinations of miRNAs with routine serum proteins such as CEA or CA-15-3 may increase the diagnostic success rate [47]. In our routine practice, we do not use these serum proteins for diagnostic purposes but for BC treatment follow-up of our patients. However, when we examined whether there was any correlation between these proteins and miRNA levels, we did not detect any correlation.

The study has some limitations. Firstly, the distribution of some characteristics of the participants may not be similar between the patient and control groups, and this difference may lead to conflicting results. The second and main limitation of this study is its relatively small sample size and single-center design, which may limit generalizability. Multicenter studies will be required to validate these findings and explore the prognostic or therapeutic value of these miRNAs.

## 4. Materials and Methods

The study was approved by the Ethics Committee of Istanbul University (Approval number: 2025/304).

### 4.1. Study Cohort

The study included 123 individuals admitted to Istanbul University Oncology Institute for BC treatment or screening between January 2023 and June 2023. Among these, 92 persons diagnosed with BC constituted our patient group with histopathology. We used the AJCC Cancer Staging Manual for staging BC patients. The luminal group contains Luminal A-B, and the non-luminal group contains HER2-positive and triple-negative groups. In comparison, 31 persons who were not diagnosed with BC and referred to the clinic for routine breast examination constituted our control group. All patients obtained written informed consent. The Ethics Committee of Istanbul University (decision number no: 2025/304) approved the study. All procedures performed in this study involving human participants followed the ethical standards of the institutional research committee and the tenets of the Helsinki Declaration of 1975, as revised in 1983 and its later amendments or comparable ethical standards. Exclusion criteria included patients with metastatic disease, persons who had previously suffered from a different cancer, and persons who had concerns about the study and did not consent to participation. We collected blood samples from BC patients undergoing initial surgical treatment before the operation and from patients referred for neoadjuvant therapy before the first neoadjuvant regimen. Peripheral venous blood in gel biochemistry tubes (BD Vacutainer, SST II, BD Company, Franklin Lakes, NJ, USA) was collected from the patients and healthy individuals to be included in the study. Then, it was kept at room temperature for 30 min for clotting and centrifuged (Hettich Universal 32, Tuttlingen, Germany) at 3000 rpm for 10 min. After centrifugation, the upper serum fluid was transferred to microtubes. There were no hemolyzed samples among the samples, and when there were, we did not use them directly. In such cases, we took new samples if possible.

### 4.2. Quality Control Measurements of Biological Material

RNA purity was assessed by spectrophotometric measurements of A260/A280 and A260/A230 ratios. The A260/A280 ratio is a standard indicator for detecting protein and phenolic contamination in RNA samples, with ideal values typically ranging between 1.8 and 2.1. The A260/A230 ratio indicates the presence of organic solvent residues, salts, and other contaminants; high-quality RNA samples usually exhibit values above 2.0. The presented samples showed A260/A280 ratios ranging from 2.03 to 2.17, suggesting minimal protein contamination. The A260/A230 ratios varied between 0.93 and 2.23, indicating low levels of organic contamination in some samples. RNA concentrations confirm that a sufficient quantity and quality of RNA were obtained for downstream molecular analyses. These quality control parameters support the suitability of the RNA samples for reliable results in subsequent applications such as qRT-PCR and next-generation sequencing (NGS).

### 4.3. Determination of miRNA Expressions

Total RNA was isolated from serum samples using Qiazol (Qiagen, Cat No: 79306, Dusseldorf, Germany), a guanidium thiocyanate solution. The concentrations of the isolated RNA samples were subsequently equalized before complementary DNA (cDNA) synthesis using the miRCURY LNA RT Kit (Qiagen, Catalog No: 339340, Dusseldorf, Germany). For each sample, a 10 µL mixture containing 2 µL of 5× miRCURY RT Reaction Buffer, 4.5 µL of RNase-free water, 1 µL of 10× miRCURY RT Enzyme Mix, 0.5 µL of synthetic RNA spikeins, and 2 µL of template RNA was added to 0.2 µL tubes and vortexed. The mixture was incubated in a C1000 Touch Thermal Cycler (Bio-Rad, Hercules, CA, USA) at 42 °C for 60 min, followed by a 5 min incubation at 95 °C to inactivate the reaction. The cDNA was subsequently stored at −20 °C until the subsequent experimental step. Following cDNA synthesis, the expression levels of miR-197-3p, miR-1236, and miR-1271 were quantified by the qRT-PCR system, CFX96 Touch (Bio-Rad, Hercules, CA, USA), employing specific validated miRNA primers (Qiagen, Dusseldorf, Germany). The miRCURY LNA SYBR Green PCR Kit (Qiagen, Catalog No: 339345, Dusseldorf, Germany) was used to determine miRNA expression levels according to the manufacturer’s protocol. The RNU6 gene was utilized as a housekeeping control, as it is the most widely used endogenous gene [48]. The same persons’ samples were analyzed for all three microRNAs (miR-197-3p, miR-1236, and miR-1271) and the respective endogenous control (RNU6) on the same plate and as two replicates.

In the analysis of qRT-PCR data, relative expression was calculated by comparing the PCR signal of the target region in the patient group to the PCR signal in the control group. Changes in Ct values obtained from the device were analyzed using the ΔΔCt method. Normalization calculations of expression analysis of miRNA genes were performed with housekeeping RNU6 using the following formula:

ΔCt = Ct (target gene) − Ct (reference gene), ΔΔCt = mean patients’ ΔCt − mean ΔCt of controls. The change in the expression level of a studied gene compared to the control group is called a Fold Change (FC) value. The expression level alternations of a gene were calculated with the formula FC = 2^−ΔΔCt^ [49].

### 4.4. Statistical Analysis

The sample size was calculated with the G*Power Version 3.1.9 program. For the miRNA ΔCt values measured between BC patients and healthy controls, the difference in the medium effect size (effect size = 0.5) was predicted to be statistically significant, and the sample size was determined as a minimum of 45 participants for 95% power at a 0.05 alpha significance level.

Statistical analyses were conducted using SPSS Software (version 22, Chicago, IL, USA). To assess the normality of the variables, analytical methods such as the Kolmogorov–Smirnov/Shapiro–Wilk’s test were employed. Since the variables did not meet the parametric assumptions, the non-parametric Mann–Whitney U test determined statistical significance between parameters. Additionally, the chi-square test was utilized to evaluate whether the observed frequencies deviated significantly from the expected frequencies. A *p*-value less than 0.05 was deemed statistically significant. In response to these significant findings, the Bonferroni method within the false discovery rate (FDR) framework for multiple testing correction was further applied, strictly setting the significance threshold at *p* < 0.05. Furthermore, the diagnostic values of the tests were analyzed using the ROC (Receiver Operating Characteristic) curve.

## 5. Conclusions

In summary, our results demonstrated that miR-197-3p was significantly upregulated and miR-1236 was markedly downregulated in BC patients compared to healthy controls, while miR-1271 showed no significant difference. The upregulation of miR-197-3p aligns with existing evidence suggesting its oncogenic role in breast and other solid tumors, where it promotes proliferation and inhibits apoptosis. Conversely, miR-1236 has been described as a tumor suppressor, and our findings confirm its reduced expression in BC patients, consistent with studies linking low levels to increased tumor invasion and poor prognosis. Serum miR-197-3p and miR-1236 show promise as non-invasive biomarkers for early BC detection. Their combined use enhances diagnostic power and may contribute to future liquid biopsy strategies in clinical oncology.

As seen in many previous studies, miRNAs can be potential diagnostic and prognostic biomarker targets. To elucidate the detailed functions of the studied miRNAs, miR-197-3p and miR-1236, identify target genes and signaling pathways, determine treatment response relationships, examine their distinct roles according to tumor subtypes and stages, and perform survival analyses, larger patient groups and multicenter studies are required. In addition, further research with larger case series is essential to investigate the functional roles of these microRNAs in tumor progression and their potential therapeutic applications.

## Figures and Tables

**Figure 1 ijms-26-08944-f001:**
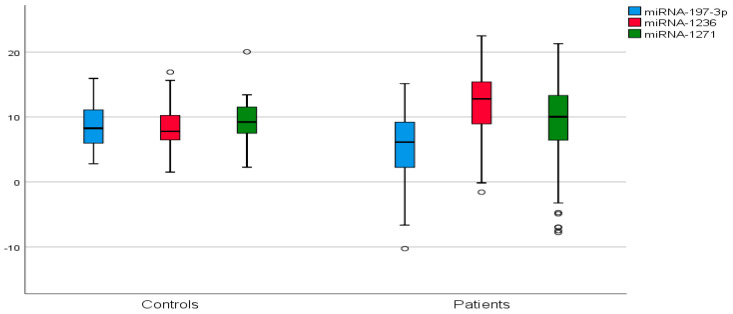
Comparison of serum ΔCt values of miR-197-3p, miR-1236, and miR-1271 between breast cancer patients and healthy controls. Box plots illustrate median, interquartile ranges, and minimum–maximum ΔCt values for each miRNA in breast cancer patients and controls. Lower ΔCt values indicate higher miRNA expression. *p* < 0.05 indicates statistically significant differences.

**Figure 2 ijms-26-08944-f002:**
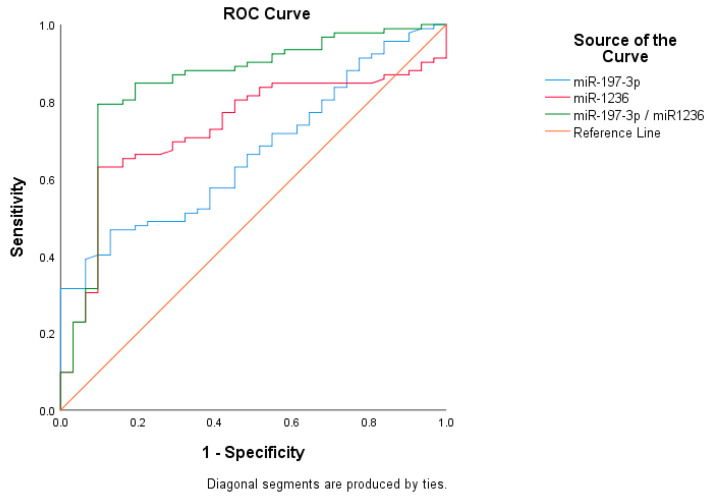
Receiver operating characteristic (ROC) curves of miR-197-3p, miR-1236, and their combination in breast cancer diagnosis. ROC curves illustrate the diagnostic performance of serum miR-197-3p, miR-1236, and their combined use. The combined biomarker model demonstrated the highest area under the curve (AUC = 0.842), indicating superior discriminatory ability between breast cancer patients and controls.

**Table 1 ijms-26-08944-t001:** Serum ΔCt values and fold change coefficients of miR-197-3p, miR-1236, and miR-1271 in breast cancer patients and healthy controls.

miRNA	ΔCt		STD		2^−ΔΔCt^		Fold Change	*p*
	Patients	Controls	Patients	Controls	Patients	Controls		
miR-197-3p	5.32	8.48	5.31	3.47	0.02503	0.00280	8.939	**0.0048**
miR-1236	11.66	8.49	5.08	3.23	0.00031	0.00278	0.112	**0.0029**
miR-1271	8.69	9.20	6.29	3.63	0.00242	0.001710	1.42	0.6712

ΔCt values and fold change (2^−ΔΔCt^) indicate relative expression levels of circulating serum miRNAs in breast cancer patients versus healthy controls. Significant *p*-values (<0.05) are shown in bold.

**Table 2 ijms-26-08944-t002:** Univariate and multivariate logistic regression analyses of miRNA expression and patient characteristics between breast cancer patients and controls.

	Univariate Model				Regression Model		
		Patients	Controls	*p*	OR	95%CI Exp(B)	*p*
miR-197-3p	Low	29 (31.5%)	18 (58.1%)	0.008	0.33	0.14–0.77	0.010
	High	63 (68.5%)	13 (41.9%)				
miR-1236	Low	65 (70.7%)	10 (32.3%)	<0.001	5.06	2.10–12.15	<0.001
	High	27 (29.3%)	21 (67.7%)				
miR-1271	Low	36 (39.1%)	13 (41.9%)	0.472	1.12	0.49–2.59	0.783
	High	56 (60.9%)	18 (58.1%)				
Age	<50	43 (46.7%)	19 (61.3%)	0.161	1.80	0.77–4.14	0.164
	>50	49 (53.3%)	12 (38.7%)				
Menopause	Pre	42 (45.7%)	20 (64.5%)	0.069	2.17	0.93–5.03	0.072
	Post	50 (54.3%)	11 (35.5%)				
Birth history	No	13 (14.1%)	15 (48.4%)	<0.001	5.70	2.28–14.25	<0.001
	Yes	79 (85.9%)	16 (51.6%)				
Bad habits	No	84 (91.3%)	19 (61.3%)	<0.001	0.15	0.05–0.42	<0.001
	Yes	8 (8.7%)	12 (38.7%)				
Oral contraceptive	No	86 (93.5%)	18 (58.1%)	<0.001	0.09	0.03–0.29	<0.001
	Yes	6 (6.5%)	13 (41.9%)				

**Table 3 ijms-26-08944-t003:** ROC curve analysis outcomes of miR-197-3p and miR-1236 for breast cancer diagnosis.

Assay	Cut-Off Value	Sensitivity (%)	Specificity (%)	PPV (%)	NPV (%)	AUC (CI)	*p*
miR-197-3p	7.32	67.0%	45.6%	79.6%	30.9%	0.667 (0.567–0.786)	0.0056
miR-1236	9.87	80.7%	51.3%	89.3%	53.7%	0.731 (0.636–0.827)	0.0001

AUC: area under the curve; PPV: positive predictive value; NPV: negative predictive value.

**Table 4 ijms-26-08944-t004:** ROC curve analysis of combined miR-197-3p and miR-1236 for breast cancer diagnosis.

			Asymptotic 95% Confidence Interval	
Test Variable	AUC	Asymptotic Sig	Lower Bound	Upper Bound
miR-197-3p–miR-1236	0.842	<0.001	0.764	0.936

**Table 5 ijms-26-08944-t005:** miR-197-3p, miR-1236 levels, and tumor features.

			miR-197-3p		miR-1236	
Variables		*n* (%)	Mean ± STD	*p*	Mean ± STD	*p*
Age	<50	35 (38.0%)	5.68 ± 4.39	0.053	12.32 ± 3.59	0.326
	≥50	57 (62.0%)	7.62 ± 4.48		11.25 ± 5.80	
Tumor size	<2 cm	36 (39.1%)	4.86 ± 5.34	0.510	11.17 ± 3.58	0.463
	≥2 cm	56 (60.9%)	5.61 ± 5.32		11.97 ± 4.75	
Breast cancer subtype	IDC	75 (81.5%)	5.59 ± 5.13	0.303	11.57 ± 4.95	0.733
	Other BC subtypes *	17 (18.5%)	4.11 ± 6.04		12.04 ± 5.76	
Histologic grade	1–2	59 (64.1%)	5.58 ± 5.07	0.531	11.84 ± 5.15	0.647
	3	33 (35.9%)	4.85 ± 5.77		11.33 ± 5.01	
cT	1–2	65 (70.7%)	5.24 ± 5.02	0.842	11.53 ± 5.13	0.716
	3	27 (29.3%)	5.50 ± 5.25		11.85 ± 5.02	
cN	Negative	52 (56.5%)	5.86 ± 5.07	0.265	11.82 ± 5.27	0.723
	Positive	40 (43.5%)	4.81 ± 5.59		11.44 ± 4.88	
ER	Negative	15 (16.3%)	5.33 ± 5.14	0.891	11.95 ± 4.91	0.806
	Positive	77 (83.7%)	5.31 ± 5.37		11.60 ± 5.14	
PR	Negative	20 (21.7%)	4.96 ± 4.97	0.737	11.06 ± 4.91	0.554
	Positive	72 (78.3%)	5.41 ± 5.43		11.82 ± 5.15	
HER2	Negative	72 (78.3%)	5.48 ± 5.16	0.569	11.97 ± 4.84	0.261
	Positive	20 (21.7%)	4.71 ± 5.91		10.52 ± 5.84	
Receptor status	Luminal	76 (82.6%)	5.38 ± 5.37	0.807	11.66 ± 5.13	0.956
	Non-luminal	16 (17.4%)	5.02 ± 5.12		11.59 ± 4.96	
Tumor stage	1–2	79 (85.9%)	5.14 ± 5.41	0.439	11.53 ± 5.13	0.718
	3	13 (14.1%)	6.38 ± 4.72		11.95 ± 5.02	

* Other breast cancer subtypes include invasive lobular carcinoma, invasive mucinous carcinoma, and tubular carcinoma. IDC: invasive ductal carcinoma; BC: breast cancer; cT: clinical stage for tumor size; cN: clinical stage for lymph nodes; ER: estrogen receptor positivity; PR: progesterone receptor positivity; HER2: human epidermal growth factor receptor.

## Data Availability

The original contributions presented in this study are included in the article. Further inquiries can be directed to the corresponding author.

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
