# Peer review of "Exploratory Analysis of Circulating Serum miR-197-3p, miR-1236, and miR-1271 Expression in Early Breast Cancer"

_ijms, 2025, doi:10.3390/ijms26188944_

Round 1

Reviewer 1 Report

Comments and Suggestions for Authors

In the manuscript, the authors have studied the diagnostic role of circulating miRNAs in the early detection of breast cancer (BC). The serum samples from BC patients and healthy people were sampled and the miRNA levels were quantified by qRT-PCR. The results suggest an upregulation of miR-197-3p and a down regulation in miR-1236 expression levels in the tested serum samples of the BC patients relative to healthy control group. The manuscript is well written. Mentioned below are comments.

Comments

  1. Introduction section, line 97: “miR-197-3p is a small RNA molecule containing an oligonucleotide sequence at the 5' end” It is inaccurate to mention that the oligonucleotide sequence is present at the 5’ end. miR-197-3p is a 22-nucleotide long sequence.

  1. In the introduction section, line 7 comma is missing. The sentence mentions 43.000 instead of 43,000.

  1. Y-axis legend is missing in figure 1 and figure 2.

  1. Table 6 includes abbreviations like cT, CN, ER, PR, IDC. Kindly mention their full forms in the table caption.

Author Response

C1: Introduction section, line 97: “miR-197-3p is a small RNA molecule containing an oligonucleotide sequence at the 5' end” It is inaccurate to mention that the oligonucleotide sequence is present at the 5’ end. miR-197-3p is a 22-nucleotide long sequence.

A1: You're absolutely right about this, we did correction.

C2: In the introduction section, line 7 comma is missing. The sentence mentions 43.000 instead of 43,000.

A2: We did your recommendation.

C3: Y-axis legend is missing in figure 1 and figure 2.

A3: There can't be a y-axis in tables. These are comparisons within groups. That's why we didn't include a y-axis.

C4: Table 6 includes abbreviations like cT, CN, ER, PR, IDC. Kindly mention their full forms in the table caption.

A4: We put the full names.

Reviewer 2 Report

Comments and Suggestions for Authors

This manuscript investigates the potential diagnostic utility of three circulating serum miRNAs (miR-197-3p, miR-1236, and miR-1271) for early breast cancer (BC) detection using qRT-PCR. The authors report significant upregulation of miR-197-3p and downregulation of miR-1236 in BC patients. The topic is timely and relevant to the field of non-invasive cancer biomarkers, especially in breast oncology. However, the manuscript requires major revisions before it can be considered for publication.

Major comments:

  • The title does not fully reflect the nature of the work. The study only measures circulating levels of miR-197-3p, miR-1236, and miR-1271 in patients and controls. No functional experiments or in silico analyses were performed to investigate whether these miRNAs have an impact biological role in BC development. Their involvement in tumorigenesis remains speculative; only functional validation will confirm whether the miRNAs directly contribute to tumor development or whether they are merely correlative markers. A more accurate title should reflect that this is an observational study focused on expression levels and diagnostic potential, not on biological function.

  • Lines 97–125: Please simplify the explanation of the role of each microRNA. Wouldn’t these details make more sense in the Discussion section?
  • Study cohort: It is a retrospective study, right? Indicate it in the first sentence, please.

  • Line 134-136: The manuscript states that a total of 132 individuals were enrolled in the study. However, only 92 BC patients and 31 healthy controls are accounted for in the results, totaling 123 participants. Please clarify

  • How were the patients diagnosed with BC? Histopathology and immunophenotype? Please clarify the methods used for confirming the diagnosis. Furthermore, in Table 6, the authors present data on breast cancer type, histological grade, receptor status, and tumor stage. However, the manuscript does not describe how this information was obtained, or which classification systems were used. Please specify and use the references for support. Additionally, the non-luminal subtypes were defined as HER2-overexpressing and triple-negative/basal-like? What scoring system was used to define HER2 positivity (+3 only)?

  • Exclusion criteria: Patients with concurrent tumors were considered as an exclusion criterion?

  • Line 145-147: Please provide a more detailed explanation of how the blood samples were collected. Was a standard venipuncture method used? Kindly specify the types of tubes used for blood collection and describe the centrifugation process employed to separate the fractions.

  • Hemolysis is a critical pre-analytical factor in circulating miRNA studies, as it can artificially alter miRNA expression profile. The manuscript does not mention whether hemolysis control was performed on the serum samples prior to RNA extraction. Please clarify

  • Quality control measurements of biological material: Table 1 reports RNA quality assessment parameters (A260/280, A260/230, and RNA concentration) for only eight samples, despite the total cohort comprising 132 participants. Were RNA quality assessments performed on these eight samples? Additionally, the table does not specify whether these samples correspond to BC patients or healthy controls. Ensuring RNA purity and integrity across all samples is essential, particularly in serum-based qRT-PCR studies, where RNA degradation can significantly affect results.

  • It is extremely necessary to explain why these microRNAs were selected. Furthermore, the nomenclature of the microRNAs should be clarified. For instance, miR-197-3p specifies the 3’ arm of the precursor, while miR-1236 lacks this indication. Please explain why some microRNAs are annotated with the arm designation (3p or 5p) and others are not.

  • Please clarify why RNU6 was used as the endogenous control gene. Please support this with references.

  • The same patients' samples were analyzed for the three microRNAs and respective endogenous control on the same plate. How many replicates were performed on the same plate for each sample?

  • Should Figure 2 not follow the same format as Figure 1 for consistency? Additionally, given that the data are non-parametric, would it not be more appropriate to display variability using interquartile ranges or minimum and maximum values.

  • Line 310 – 329: It is understandable that the authors wish to provide context and cite examples of other microRNAs with well-established functions in the literature before stating the objective in the discussion. However, the exposition of the evidence presented seems rather extensive for the sole purpose of contextualizing the objective.

  • The authors correctly acknowledge in the discussion that the individual AUC values for each microRNA are relatively low. However, when combined, the AUC reaches 0.842, and they suggest that this combination may serve as a potential biomarker. While the improvement is notable, is it not somewhat speculative to propose clinical utility with an AUC still below 0.9?

  • The study's limitations and future perspectives should be addressed in greater depth. a) For example, investigating the functional roles of these miRNAs in tumor progression and their potential therapeutic applications would enhance the manuscript's clinical relevance. b) Have these microRNAs been previously studied in animals? If not, this could also serve as a good starting point for comparative oncology. The “One Health” approach has been gaining increasing attention (please see https://doi.org/10.3390/life12040524 and https://doi.org/10.1007/s11357-024-01260-7).

Minor comments:

  • Line 39: Please, remove extra parenthesis in “specificity = 45.6%))”.
  • Line 82: “Consequently, Most …” Please change to most.
  • Line 132: Note that the sentence is missing a closing parenthesis.
  • Material and methos: Please specify the assay IDs of the primers used.
  • Line 219: “Table 2, Figure 1”. Please change to “Table 2 and Figure 1”.
  • Table 2: Please remove the bold formatting from the value 5.32.
  • Figure 1: please add caption axis.
  • Figure 1: Is the bar corresponding to miR-1236 correct?
  • Results: Please remove the word “legend” from the footnotes of tables and figures. For figures, merge the legend directly with the figure caption.
  • The quality of Figure 1 should be improved.
  • Table 6: Please include the abbreviation IDC, cT and Cn in a footnote. For example, “IDC: Invasive Ductal Carcinoma”.
  • Line 299: Please remove “microRNA,” as the abbreviation has already been defined earlier.
  • Line 302 and 305: “Recent studies…” Please cite the relevant studies within this sentence.
  • Discussion: There is a paragraph in the Discussion section that is incorrectly formatted in bold. Please adjust the formatting accordingly.
  • Line 355: Is it mir-197-3p or miR-197-3p?
  • Line 358-359: “Studies in BC cells show 358 that the expression of miR-197-3p is usually high”. Please add the references related to the studies that authors mention.
  • Line 413: Please change “studies are” to “one study showed”.
  • Please explain what “STD” refers to.
  • Line 418: “Some studies….”. Please add references related to the studies.
  • Line 430 and 433: Please change “review” to “meta-analysis”
  • Although the conclusion is currently integrated into the discussion, the addition of a separate section entitled “Conclusion” could enrich the manuscript by providing a clearer and more accessible summary of the main findings for readers. However, as the journal's guidelines state that this section is optional, this suggestion is left to the discretion of the authors.

Author Response

Major comments:

  • The title does not fully reflect the nature of the work. The study only measures circulating levels of miR-197-3p, miR-1236, and miR-1271 in patients and controls. No functional experiments or in silico analyses were performed to investigate whether these miRNAs have an impact biological role in BC development. Their involvement in tumorigenesis remains speculative; only functional validation will confirm whether the miRNAs directly contribute to tumor development or whether they are merely correlative markers. A more accurate title should reflect that this is an observational study focused on expression levels and diagnostic potential, not on biological function.
  • You're absolutely right about this. Our goal was to demonstrate whether these markers could be used for early detection of breast cancer in the future. Larger studies, particularly on biological factors, are needed on this topic.

  • Lines 97–125: Please simplify the explanation of the role of each microRNA. Wouldn’t these details make more sense in the Discussion section?
  • You are right, information that is not necessary for this tab has been restricted.
  • Study cohort: It is a retrospective study, right? Indicate it in the first sentence, please.
  • İt is not a retrospective study. We collect the materials and when we finish collecting we start to study samples. Who study samples never know anything about the stage or other clinical information about the patients.
  • Line 134-136: The manuscript states that a total of 132 individuals were enrolled in the study. However, only 92 BC patients and 31 healthy controls are accounted for in the results, totaling 123 participants. Please clarify
  • We change it.
  • How were the patients diagnosed with BC? Histopathology and immunophenotype? Please clarify the methods used for confirming the diagnosis. Furthermore, in Table 6, the authors present data on breast cancer type, histological grade, receptor status, and tumor stage. However, the manuscript does not describe how this information was obtained, or which classification systems were used. Please specify and use the references for support. Additionally, the non-luminal subtypes were defined as HER2-overexpressing and triple-negative/basal-like? What scoring system was used to define HER2 positivity (+3 only)?
  • We diagnosed BC patients with Histopathology. We add this material and method part. We use AJCC Cancer Staging Manual. We add the explanations. Luminal group is Luminal A-B and Non Luminal Group is HER 2 positive and triple negative group. We add this explanations to our manuscript.
  • Exclusion criteria: Patients with concurrent tumors were considered as an exclusion criterion?
  • Yes.

  • Line 145-147: Please provide a more detailed explanation of how the blood samples were collected. Was a standard venipuncture method used? Kindly specify the types of tubes used for blood collection and describe the centrifugation process employed to separate the fractions.
  • Peripheral venous blood in gel biochemistry (yellow-capped) tubes was collected from the patients and healthy individuals to be included in the study. Then, it was kept at room temperature for 30 min for clotting and centrifuged at 3000 rpm for 10 min. After centrifugation, the upper serum fluid was transferred to ependorf tubes. 
  • Hemolysis is a critical pre-analytical factor in circulating miRNA studies, as it can artificially alter miRNA expression profile. The manuscript does not mention whether hemolysis control was performed on the serum samples prior to RNA extraction. Please clarify
  • We knew that hemolysis was a critical pre-analytical factor, and there were no hemolyzed samples among the samples, so we did not use them directly. In such cases, we take new samples if possible. Added in the section

  • Quality control measurements of biological material: Table 1 reports RNA quality assessment parameters (A260/280, A260/230, and RNA concentration) for only eight samples, despite the total cohort comprising 132 participants. Were RNA quality assessments performed on these eight samples? Additionally, the table does not specify whether these samples correspond to BC patients or healthy controls. Ensuring RNA purity and integrity across all samples is essential, particularly in serum-based qRT-PCR studies, where RNA degradation can significantly affect results.
  • Before moving on to the actual experiments, we optimize the RNA isolation methods for the most optimal conditions and then proceed to the actual experiments.
  • It is extremely necessary to explain why these microRNAs were selected. Furthermore, the nomenclature of the microRNAs should be clarified. For instance, miR-197-3p specifies the 3’ arm of the precursor, while miR-1236 lacks this indication. Please explain why some microRNAs are annotated with the arm designation (3p or 5p) and others are not.
  • Those without arms become precursor miRNAs. Those with 5p- or 3p- are usually mature miRNAs. As you know, all kinds of studies can be conducted, so we based this study on information from the literature.
  • Please clarify why RNU6 was used as the endogenous control gene. Please support this with references.
  • We used miRNA because it is one of the most widely used endogenous genes in miRNA expression studies. Reference added.
  • The same patients' samples were analyzed for the three microRNAs and respective endogenous control on the same plate. How many replicates were performed on the same plate for each sample?
  • It was carried out in 2 replicates.
  • Should Figure 2 not follow the same format as Figure 1 for consistency? Additionally, given that the data are non-parametric, would it not be more appropriate to display variability using interquartile ranges or minimum and maximum values.
  • We decided that the content of Figure 2 was unnecessary as it was explained in the results section, and we removed it as per your suggestion.
  • Line 310 – 329: It is understandable that the authors wish to provide context and cite examples of other microRNAs with well-established functions in the literature before stating the objective in the discussion. However, the exposition of the evidence presented seems rather extensive for the sole purpose of contextualizing the objective.
  • You are right, we have limited the elements that do not reflect the main objective in line with your suggestion.
  • The authors correctly acknowledge in the discussion that the individual AUC values for each microRNA are relatively low. However, when combined, the AUC reaches 0.842, and they suggest that this combination may serve as a potential biomarker. While the improvement is notable, is it not somewhat speculative to propose clinical utility with an AUC still below 0.9?
  • Yes, it's debatable. We acknowledge that larger series are needed to fully disclaim its use for this purpose. But we must also remember that we are on an acceptable path.

  • The study's limitations and future perspectives should be addressed in greater depth. a) For example, investigating the functional roles of these miRNAs in tumor progression and their potential therapeutic applications would enhance the manuscript's clinical relevance. b) Have these microRNAs been previously studied in animals? If not, this could also serve as a good starting point for comparative oncology. The “One Health” approach has been gaining increasing attention (please see https://doi.org/10.3390/life12040524 and https://doi.org/10.1007/s11357-024-01260-7).
  • Investigating the functional roles of these microRNAs in tumor progression and their potential therapeutic applications has guided our research, guided by this article. We've found exciting results in this area, and we're eager to share them, so we're publishing this study. We eagerly await the publication of our article and the contributions it provides, so we can further improve our work. To our knowledge, there are no studies conducted on animals. Our goal is to improve our work with suggestions like these.

Minor comments:

  • Line 39: Please, remove extra parenthesis in “specificity = 45.6%))”.
    • We did it.
  • Line 82: “Consequently, Most …” Please change to most.
    • We did it.
  • Line 132: Note that the sentence is missing a closing parenthesis.
    • We did it.
  • Material and methos: Please specify the assay IDs of the primers used.
  • Line 219: “Table 2, Figure 1”. Please change to “Table 2 and Figure 1”.
    • We did it.
  • Table 2: Please remove the bold formatting from the value 5.32.
    • We did it.
  • Figure 1: please add caption axis.
    • We did it.
  • Figure 1: Is the bar corresponding to miR-1236 correct?
    • yes
  • Results: Please remove the word “legend” from the footnotes of tables and figures. For figures, merge the legend directly with the figure caption.
    • We did it.
  • The quality of Figure 1 should be improved.
  • Table 6: Please include the abbreviation IDC, cT and Cn in a footnote. For example, “IDC: Invasive Ductal Carcinoma”.
    • We did it.
  • Line 299: Please remove “microRNA,” as the abbreviation has already been defined earlier.
    • We did it.
  • Line 302 and 305: “Recent studies…” Please cite the relevant studies within this sentence.
    • We did it.
  • Discussion: There is a paragraph in the Discussion section that is incorrectly formatted in bold. Please adjust the formatting accordingly.
    • We did it.
  • Line 355: Is it mir-197-3p or miR-197-3p?
    • İt is miR-197-3p. We changed it.
  • Line 358-359: “Studies in BC cells show 358 that the expression of miR-197-3p is usually high”. Please add the references related to the studies that authors mention.
    • We did it.
  • Line 413: Please change “studies are” to “one study showed”.
    • We changed it.
  • Please explain what “STD” refers to.
    • We changed it.
  • Line 418: “Some studies….”. Please add references related to the studies.
    • We added the other studies.
  • Line 430 and 433: Please change “review” to “meta-analysis”
    • We did it.
  • Although the conclusion is currently integrated into the discussion, the addition of a separate section entitled “Conclusion” could enrich the manuscript by providing a clearer and more accessible summary of the main findings for readers. However, as the journal's guidelines state that this section is optional, this suggestion is left to the discretion of the authors.
    • We agree with you, but we did it this way because that is the journal writing rules.

Round 2

Reviewer 2 Report

Comments and Suggestions for Authors
  • The authors had previously agreed to modify the manuscript title; however, it remains unchanged. Please ensure that the agreed revision is implemented.
  • Line 144: The term “yellow-capped” should be removed.
  • Line 147: Please replace “ependorf tubes” with “microtubes.”
  • It is not clear whether the same patients’ samples were analyzed for all three microRNAs and the respective endogenous control on the same plate. Additionally, were two technical replicates performed for each miRNA? This methodological detail is essential and should be clarified and explicitly included in the Materials and Methods section.
  • The manuscript would benefit from the addition of a Future Perspectives section to strengthen its discussion and provide a forward-looking contribution.

Author Response

Q1: The authors had previously agreed to modify the manuscript title; however, it remains unchanged. Please ensure that the agreed revision is implemented.

A1: It should be reiterated that our study is a preliminary study and should be supported by larger prospective studies. However, it is still valuable because we have obtained striking results. When we add new and larger studies to confirm our findings, more detailed analyses and relevant visuals will be possible. Therefore, the title has been updated in line with your observation and suggestion.

Q2: Line 144: The term “yellow-capped” should be removed.

A2: We did it.

Q3: Line 147: Please replace “ependorf tubes” with “microtubes.”

A3: We did it.

Q4: It is not clear whether the same patients’ samples were analyzed for all three microRNAs and the respective endogenous control on the same plate. Additionally, were two technical replicates performed for each miRNA? This methodological detail is essential and should be clarified and explicitly included in the Materials and Methods section.

A4: We did it.

Q5: The manuscript would benefit from the addition of a Future Perspectives section to strengthen its discussion and provide a forward-looking contribution.

A5: The addition you highlighted has been added to the discussion section.

Although many parts of our article, especially the English grammar editing section, have been completed, we apologize for the delays in editing our article due to some technical difficulties at our institute.

Round 3

Reviewer 2 Report

Comments and Suggestions for Authors

-The control group (n = 31) is substantially smaller than the patient group (n = 92). Could this uneven distribution reduce statistical power and bias the results, particularly for the ROC analysis and logistic regression?

-Line 532 – “suitability of the RNA samples for…NGS”

The mention of next-generation sequencing is irrelevant since no NGS experiments were performed in this study.

- Methods Section - RNA Quality: The authors note that the A260/A230 ratios varied between 0.93 and 2.23, indicating "low levels of organic contamination in some samples". It would be beneficial to specify how many samples had values below the ideal threshold and if this variability might have introduced any bias into the qRT-PCR results

-Table 5 – Category “Others” under breast cancer type

The label “Others*” is confusing. The footnote references invasive lobular, mucinous, and tubular carcinoma, but all are still subtypes of breast cancer. Please clarify

- Line 168 – “guanidium thiocyanate solution”

Isn't the correct term guanidine thiocyanate? Please correct as necessary.

Author Response

C1: The control group (n = 31) is substantially smaller than the patient group (n = 92). Could this uneven distribution reduce statistical power and bias the results, particularly for the ROC analysis and logistic regression?

R1: We are aware of the difference between the two groups. All patients who agreed to participate within the planned study period were included. At the end of our study, we stated that our findings were preliminary and that larger series were needed. We intend to continue collecting cases for larger series and further develop our study. Meanwhile, we want to publish these results to contribute to the literature and improve our work.

C2: Line 532 – “suitability of the RNA samples for…NGS”

The mention of next-generation sequencing is irrelevant since no NGS experiments were performed in this study.

R2: You’re right. NGS has been used as a general term. It has been added to qRT-PCR.

C3: Methods Section - RNA Quality: The authors note that the A260/A230 ratios varied between 0.93 and 2.23, indicating "low levels of organic contamination in some samples". It would be beneficial to specify how many samples had values below the ideal threshold and if this variability might have introduced any bias into the qRT-PCR results

R3: You’re right. Only 4 samples were below the ideal threshold. This did not cause any bias in the qRT-PCR results.

C4: Table 5 – Category “Others” under breast cancer type

The label “Others*” is confusing. The footnote references invasive lobular, mucinous, and tubular carcinoma, but all are still subtypes of breast cancer. Please clarify

R4: The term "others" refers to those that are not actually invasive ductal carcinoma. These include cases of invasive lobular carcinoma, mucinous carcinoma, and tubular carcinoma. We corrected in the table.

C5: Line 168 – “guanidium thiocyanate solution”

Isn't the correct term guanidine thiocyanate? Please correct as necessary

R5: Both the manufacturer's protocol and the literature indicate that the correct form is guanidium thiocyanate, which is the protonated form found in aqueous solutions.